# Effects of Energy Delivery Guided by Indirect Calorimetry in Critically Ill Patients: A Systematic Review and Meta-Analysis

**DOI:** 10.3390/nu16101452

**Published:** 2024-05-11

**Authors:** Shinichi Watanabe, Hiroo Izumino, Yudai Takatani, Rie Tsutsumi, Takahiro Suzuki, Hiroomi Tatsumi, Ryo Yamamoto, Takeaki Sato, Tomoka Miyagi, Isao Miyajima, Kensuke Nakamura, Naoki Higashibeppu, Joji Kotani

**Affiliations:** 1Department of Physical Therapy, Faculty of Rehabilitation, Gifu University of Health Science, 2-92 Higashiuzura, Gifu 500-8281, Japan; s-watanabe@gifuhoken.ac.jp; 2Acute and Critical Care Center, Nagasaki University Hospital, 1-7-1 Sakamoto, Nagasaki 852-8501, Japan; izumino@nagasaki-u.ac.jp; 3Department of Primary Care and Emergency Medicine, Kyoto University Hospital, 54 Shogoin-Kawahara-cho, Sakyo-ku, Kyoto 606-8507, Japan; takataniyu@kuhp.kyoto-u.ac.jp; 4Department of Nutrition and Metabolism, Institute of Biomedical Sciences, Tokushima University Graduate School, 3-18-15 Kuramoto, Tokushima 770-8503, Japan; rtsutsumi@tokushima-u.ac.jp; 5Department of Cardiovascular Medicine, St. Luke’s International Hospital, 9-1 Akashi-cho, Chuo-ku, Tokyo 104-8560, Japan; takasu.623@gmail.com; 6Department of Intensive Care Medicine, Sapporo Medical University School of Medicine, S1 W17, Chuo-ku, Sapporo 060-8556, Japan; htatsumi@sapmed.ac.jp; 7Department of Emergency and Critical Care Medicine, Keio University School of Medicine, 35 Shinanomachi, Shinjuku-ku, Tokyo 160-8582, Japan; ryoyamamoto@keio.jp; 8Emergency Center, Tohoku University Hospital, 1-1 Seiryo-machi, Aoba-ku, Sendai 980-8574, Japan; takesato@hkg.odn.ne.jp; 9Department of Nutrition, Yokosuka General Hospital, 2-36 Uwamachi, Yokosuka 238-8567, Japan; 10mtn.3@gmail.com; 10Department of Clinical Nutrition, Chikamori Hospital, 1-1-16 Okawasuzi, Kochi 780-8522, Japan; miyajimai@y6.dion.ne.jp; 11Department of Intensive Care, Yokohama City University Hospital, 3-9 Fukuura, Kanazawa-ku, Yokohama 236-0064, Japan; mamashockpapashock@yahoo.co.jp; 12Department of Anesthesiology and Nutrition Support Team, Kobe City Medical Center General Hospital, 2-1-1 Minatojima Minamimachi, Chuo-ku, Kobe 650-0047, Japan; gashibe@hotmail.com; 13Division of Disaster and Emergency Medicine, Department of Surgery Related, Kobe University Graduate School of Medicine, 7-5-1 Kusunoki-cho, Chuo-ku, Kobe 650-0017, Japan

**Keywords:** critically ill patients, indirect calorimetry, meta-analysis

## Abstract

Background: The utility of using indirect calorimetry (IC) to estimate energy needs and methods for its application to this purpose remain unclear. This systematic review investigated whether using IC to estimate energy expenditure in critically ill patients is more meaningful for improving survival than other estimation methods. Methods: Comprehensive searches were conducted in MEDLINE using PubMed, Cochrane Central Register of Controlled Trials, and Igaku-Chuo-Zasshi up to March 2023. Results: Nine RCTs involving 1178 patients were included in the meta-analysis. The evidence obtained suggested that energy delivery by IC improved short-term mortality (risk ratio, 0.86; 95% confidence interval [CI], 0.70 to 1.06). However, the use of IC did not appear to affect the length of ICU stay (mean difference [MD], 0.86; 95% CI, −0.98 to 2.70) or the duration of mechanical ventilation (MD, 0.66; 95% CI, −0.39 to 1.72). Post hoc analyses using short-term mortality as the outcome found no significant difference by target calories in resting energy expenditure, whereas more frequent IC estimates were associated with lower short-term mortality and were more effective in mechanically ventilated patients. Conclusions: This updated meta-analysis revealed that the use of IC may improve short-term mortality in patients with critical illness and did not increase adverse events.

## 1. Introduction

The amount of energy administered to critically ill patients is important because it may affect their prognosis [1,2]. Estimation formulae using body weight, age, and height have been used to select the amount of energy to deliver [3]. Previous studies reported the effectiveness of indirect calorimetry (IC) to measure energy expenditure [4,5,6,7]. Although a number of institutions are investigating the effectiveness of IC, its feasibility varies depending on a number of factors, such as the facility size and workforce.

A systematic review (SR) previously suggested that the use of IC to guide energy delivery in critically ill patients affected clinical outcomes more than predictive equations [8,9]. However, it was limited by the small sample sizes of the studies selected for investigation as well as the lack of disease-specific sub-analyses and physical assessments. Furthermore, the timing of the initiation of IC varied among the eight studies included in the SR [9].

Several guidelines recommend using IC to estimate energy expenditure [10,11,12]. However, in the early implementation of IC, there is no standardized method for calculating the percentage of energy to deliver based on energy expenditure estimated by IC. In some countries, IC availability is limited due to uncertain IC measurement conditions, technological and cost issues limitations [13]. Moreover, a number of factors affect IC measurements, including non-mechanical ventilation and severe ventilator conditions [14].

The use of IC in ICUs has increased in recent years, and several randomized controlled trials (RCTs) have been conducted to assess its effectiveness. A single-center RCT in mechanically ventilated patients reported that nutritional therapy managed based on continuous IC measurements significantly reduced mean daily energy deficit and ICU mortality [15]. The hypothesis of this study is that using IC to estimate energy expenditure in critically ill patients will improve survival compared to other estimation methods. In this study, we performed an updated SR and meta-analysis of these RCTs to evaluate the effectiveness of using IC with multiple sub-analyses to clarify the timing of the initiation of IC and target energy depending on this timing.

## 2. Materials and Methods

This SR protocol was registered in the Open Science Framework, a non-profit technology organization based in Charlottesville, Virginia (https://www.cos.io/products/osf/OSF/, accessed on 20 April 2023; Registration DOI. https://doi.org/10.17605/OSF.IO/H7RAB, accessed on 20 April 2023). This protocol follows the Preferred Reporting Items for Systematic Review and Meta-Analysis 2020 (PRISMA-2020) [16,17,18], and the SR is reported following the PRISMA guidelines [16,19,20] (Appendix A).

### 2.1. Search Strategy

We searched the following databases for eligible full-text clinical trials conducted on humans in English or Japanese, from their inception to 31 March 2023: MEDLINE via PubMed, Cochrane Central Register of Controlled Trials, and Igaku-Chuo-Zasshi. Details on the search strategy and terms used in each database are shown in Appendix A. We examined the reference lists of studies and international guidelines (American Society for Parenteral and Enteral Nutrition Guidelines 2016 [10], European Society for Clinical Nutrition and Metabolism Guidelines 2019 [11]) as well as those of eligible studies and articles that cited these studies.

### 2.2. Data Extraction

Two independent researchers (TS and SW) screened titles and abstracts and assessed their eligibility based on their full texts. A second screening was performed to match eligibility criteria for the full manuscript. A design form was used for data extraction and included information on the study design, population characteristics, number of participants, age, Acute Physiology and Chronic Health Evaluation II score, intervention protocol (intervention duration and frequency), controls, and outcomes. Differences in screening results were resolved by discussions; if this failed, a third reviewer (HI) acted as an arbitrator.

### 2.3. Inclusion and Exclusion Criteria

Original RCTs in English or Japanese were included. Observational studies without interventions, RCT secondary analyses, and post hoc analyses were excluded. The population of interest was critically ill adult patients aged ≥ 18 years. Animal studies were excluded. The intervention of interest was nutritional administration based on energy consumption, as measured by IC. We defined the control group as nutritional administration based on energy consumption using the estimation formula. However, we did not define the intervention provider category nor the method of nutritional administration (such as the full dose and gradual escalations) based on the amount of energy measured using IC.

### 2.4. Outcomes

The primary outcome of this SR was (1) short-term mortality (defined as ICU or hospital mortality or mortality within a 90-day follow-up after admission, with the longest observation period preferred [10]), while secondary outcomes included the following: (2) length of ICU stay, whose shortening may be associated with worse mortality or improved disease status; (3) duration of mechanical ventilation; (4) all infections; (5) ventilator-associated pneumonia; (6) physical functions (including the activities of daily living, quality of life [QOL] at discharge or thereafter up to 1 year after discharge, the Barthel Index [21], functional independence measure [22], grip strength, the Medical Research Council-sum score [23], short physical performance battery [24], 6-minute walk distance [25], Medical Outcomes Study 36-Item Short Form Health Survey [26], 12-Item Short Form Health Survey [27], and EuroQOL five dimensions 5-level) [28]); (7) changes in muscle mass during hospitalization or hospital discharge (using anthropometry, echocardiography, computed tomography, and a body composition analyzer) and (8) adverse events (liver and kidney).

### 2.5. Quality Assessment

The Cochrane Collaboration’s risk of bias assessment tool [29] was used to assess the quality of the studies examined and included seven items: random sequence generation; allocation concealment; participants and personnel; blinding of outcome assessments; incomplete outcome data; selective outcome reporting; and other biases. The risk of bias was graded as ‘low risk’, ‘some concern’, and ‘high risk’. Results are presented as a risk of bias graph and a risk of bias summary.

### 2.6. Statistical Analysis

Statistical analyses were performed using Cochrane Review Manager Software (RevMan, version 5.4). When studies used difference scales to assess continuous outcomes, such as the length of ICU stay or duration of mechanical ventilation, we planned to yield the standardized mean difference (MD); otherwise, we calculated MD. Estimates were pooled using a random-effects model; the risk ratio (RR) was estimated for dichotomized outcomes, while MD or the standard MD was estimated for continuous outcomes. We evaluated statistical heterogeneity using Q and I2 statistics [30]. Heterogeneity was considered to be significant when *p* < 0.1 or I2 < 50%. We finally classified the certainty of evidence as high, moderate, low, or very low according to the grading of recommendations, assessment, development, and evaluation system [31]. The degree was downgraded by the seriousness of limitations (risk of bias), inconsistency, the indirectness of evidence, imprecision, and publication bias. We anticipated substantial, but acceptable clinical heterogeneity, and focused on statistical heterogeneity to assess inconsistency. The indirectness of evidence refers to the generalizability of findings, which was assessed based on the relevance of the population, type of intervention, comparator, or outcomes in the included studies to our research question. We evaluated imprecision based on the confidence intervals (CIs) of the pooled results and on the sample size relative to the optimal information size.

We performed subgroup analyses of the study’s primary outcomes based on the disease of interest (burns, acute respiratory distress syndrome, pancreatitis, and sepsis). As for post hoc analyses, we conducted three sub-analyses using short-term mortality as the outcome: (1) the frequency of IC measurements (daily vs. once every ≥2 days); (2) target calories (≥90%, <90% of REE); and (3) mechanical ventilation (mechanically ventilated patients vs. non-mechanically ventilated patients included).

## 3. Results

### 3.1. Search Results

The PRISMA flow chart for selecting studies to be included in this meta-analysis is shown in Figure 1. Our search strategy yielded 1289 citations, 30 of which were considered to be potentially eligible based on their abstracts. After conducting full-text reviews, we excluded 21 citations; the reasons for their exclusion are listed in Appendix A. RCT 4, 5, 15, and 32–37, which met the eligibility criteria, were included in the review.

### 3.2. Characteristics of Included Studies

The characteristics of the nine studies that met the criteria for the meta-analysis are shown in Table 1. The analysis included 1178 patients (587 and 591 in the intervention and control groups, respectively). Among the studies analyzed, 76 patients were in the USA, 587 in Germany, 50 in Israel, 199 in Denmark, 120 in Brazil, and 146 in Malaysia. The mean or median age of patients ranged between 29.2 and 83.7 years. One study included burn patients only, some studies included patients with or without mechanical ventilation, and most studies included patients who required mechanical ventilation. The time from ICU admission to the first use of IC, the frequency of measurements, the method of nutritional management based on IC results, and the method of setting calorie targets differed in each study. Appendix A shows the algorithm for the risk of bias judgment.

### 3.3. Clinical Outcomes

Seven RCTs reported on the relationship between the use of IC and short-term mortality [4,5,32,33,35,36,37]. Evidence suggested that short-term mortality was slightly reduced by using IC to estimate energy expenditure in critically ill patients (seven studies [988 participants]: RR, 0.86; 95% CI, 0.70 to 1.06; I2 = 0%) (Appendix A, Table 2, Appendix A). The overall risk of bias was categorized as ‘low’. Due to the presence of missing outcome data, unblinding in measurements, and a potential risk in the selection of the reported results (Appendix A), the certainty of evidence for these outcomes was rated as moderate.

The length of ICU stay was assessed in seven RCTs [4,5,15,32,35,36,37]. Evidence suggested that the length of ICU stay was 0.86 days longer in the two-patient group using IC to estimate energy expenditure in critically ill patients than in the control group (seven studies [1090 participants]: MD, 0.86; 95% CI, −0.98 to 2.70; I2 = 53%) (Appendix A, Table 2, Appendix A). The overall risk of bias was classified as having ‘some concern’ (Appendix A). Imprecision in the length of ICU stay was assessed as serious because of the limited sample size. Therefore, the certainty of evidence for these outcomes was rated as low.

The duration of mechanical ventilation was assessed in seven RCTs [5,15,32,34,35,36,37]. Evidence suggested that the duration of mechanical ventilation was 0.66 days longer in the patient group using IC to estimate energy expenditure in critically ill patients than in the control group. (Seven studies [1068 participants]: MD, 0.66; 95% CI, −0.39 to 1.72; I2 = 14%) (Appendix A, Table 2, Appendix A). The overall risk of bias was classified as having ‘some concern’ (Appendix A). Imprecision was assessed as serious because of the limited sample size. Consequently, the certainty of evidence for these outcomes was rated as moderate.

Four RCTs [5,32,33,35] evaluated the impact of using IC to estimate energy expenditure in critically ill patients on the occurrence of all infections. These RCTs indicated that IC did not markedly affect the occurrence of all infections (four studies [785 participants]: RR, 0.99; 95% CI, 0.51 to 1.93, I2 = 81%) (Appendix A, Table 2, Appendix A). The overall risk of bias was categorized as ‘some concern’, and we detected heterogeneity (Appendix A). Collectively, these factors downgraded the certainty of evidence to a very low level.

Four RCTs [5,32,33,35] evaluated the impact of using IC to estimate energy expenditure in critically ill patients on ventilator-associated pneumonia. The use of IC to estimate energy expenditure did not markedly affect ventilator-associated pneumonia (four studies [785 participants]: RR, 1.06; 95% CI, 0.49 to 2.28, I2 = 57%) (Appendix A, Table 2, Appendix A). The overall risk of bias was categorized as ‘some concern’, and we detected heterogeneity (Appendix A). Collectively, these factors downgraded the certainty of evidence to a low level.

Physical function data were only available for the physical component summary of the Medical Outcomes Study 36-Item Short Form Health Survey 6 months after hospital discharge. Two RCTs [35,37] evaluated the impact of using IC to estimate energy expenditure in critically ill patients on physical function. The use of IC to estimate energy expenditure did not markedly affect physical function (two studies [309 participants]: MD, −0.06; 95% CI, −6.28 to 6.15, I2 = 0%) (Appendix A, Table 2, Appendix A). The overall risk of bias was categorized as ‘some concern’, and we detected heterogeneity (Appendix A). Collectively, these factors downgraded the certainty of evidence to a moderate level. Other outcomes planned for the analysis of physical function were not analyzed due to lack of data. Changes in muscle mass during hospitalization or at hospital discharge (using anthropometry, echocardiography, computed tomography, and a body composition analyzer) were also not analyzed due to the lack of data.

Two RCTs [5,32] reported kidney and liver dysfunction as adverse events. The use of IC to estimate energy expenditure in critically ill patients did not affect the kidney (two studies [421 participants]: RR, 1.01; 95% CI, 0.77 to 1.34; I2 = 0%) (Appendix A, Table 2, Appendix A) or liver (two studies [482 participants]: RR, 1.00; 95% CI, 0.64 to 1.57; I2 = 0%) (Appendix A, Table 2, Appendix A). The overall risk of bias was categorized as ‘some concerns’ (Appendix A). Consequently, the certainty of evidence for these outcomes was rated as moderate.

### 3.4. Subgroup Analyses

The results of subgroup analyses of short-term mortality and all infections are shown in Appendix A. No significant difference was observed in short-term mortality between two disease groups of interest (burn and non-burn patients).

### 3.5. Post Hoc Analysis (Sensitivity Analysis)

The results of three sub-analyses (frequency of IC measurements, target calories, and mechanical ventilation) with short-term mortality as the outcome are shown in Table 3 and Appendix A. Regarding the frequency of IC measurements, they were performed every 2 days or more frequently in five of the nine RCTs, with better short-term mortality results being obtained with a higher estimated frequency. No significant differences were observed in target calories, with some trials failing to deliver the target calories in both the IC and control groups. The effects of IC use on short-term mortality were slightly more beneficial in patients on mechanical ventilation.

## 4. Discussion

We investigated the effects of IC use on patients with critical illnesses in this updated SR and meta-analysis. The results of a meta-analysis based on point estimations and a certainty of evidence evaluation indicated that the use of IC improved short-term mortality. Evidence suggested that IC did not markedly affect length of ICU stay, duration of mechanical ventilation, physical function, the incidence of any infections, or adverse events. The early use of IC in critically ill patients was safe with no worsening of infections or adverse events. Post hoc analyses using short-term mortality as the outcome found no significant difference by target calories in REE; however, more frequent IC estimates were associated with lower short-term mortality and were more effective in mechanically ventilated patients. The results suggest that the use of IC may be of greater benefit for unstable, critically ill patients with diurnal fluctuations or daily changes in metabolic dynamics.

Among previous SR, some had modest sample sizes and recent studies have also focused on the effects of IC [7,29,37]; however, the effects of IC on mortality, the length of ICU stay, and the duration of mechanical ventilation have been inconsistent [6,10]. Although a recent RCT showed a significant reduction in ICU mortality in the IC group, the number of patients who died was not listed and data were not extracted [15]. One possible explanation for this discrepancy is that the distribution of underlying conditions differed between the studies included, and the effect of severe sepsis may have been present [38]. Additionally, compared to previous SRs on the effects of IC, a total of three studies (all Chinese) are not included in the statistics in this SR, and four other old and new papers have been added [10]. In the present study, we conducted a subgroup analysis based on pathophysiology and a post hoc analysis focusing on IC measurement methods. Due to significant methodological variations, it was not possible to establish effectiveness.

Post hoc analyses of short-term mortality revealed no significant differences in any measures except actual dose between the IC and control groups. Regarding the actual dose, a slight difference was noted between the IC and Estimated Formula groups based on REE only; however, the amount of energy administered was lower in the estimated formula group than in the IC group in most studies. Previous studies demonstrated that the ideal dose for mortality was approximately 70% of REE [38]. The actual dosages administered were the same or slightly higher in the IC group, but were slightly lower in the estimating formula group [11,32,33,35]. Berger et al. reported that the risk of nutrient overdosage without IC creates a barrier to the administration of enteral nutrition [39]. The control group may have been hesitant to use aggressive nutritional therapy due to uncertainty regarding the calculated REE. Results may vary depending on the frequency of measurements as well as the amount of energy administered [38]. IC measurements were previously reported to be more accurate and less error-prone in mechanically ventilated patients [13], and may be more useful in mechanically ventilated patients when short-term mortality is used as an outcome.

This SR has several limitations. The number of studies included and the small size of some of the individual samples limited the scope of our results, particularly that of subgroup analyses. This also precluded sub-group analyses to identify the diseases for which IC energy delivery is optimal. Furthermore, only English and Japanese databases were searched, which may have led to language biases. Moreover, this SR was performed using only the information available in published manuscripts; therefore, the subgroup analysis may have contained misclassifications. It is not yet known how to classify subjects by disease state when performing IC or how much energy needs to be administered based on energy calculated by IC. Finally, previous studies mentioned errors due to differences in IC devices, and the use of different IC equipment in each study may have affected the results [40]. Future studies need to investigate the potential benefits and risks of IC for energy delivery by dividing subjects according to disease states in order to establish the appropriate timing and frequency of IC measurements as well as actual calories to be administered.

## 5. Conclusions

Estimations of energy expenditure by performing IC in critically ill patients may not affect the length of ICU stay or duration of mechanical ventilation, but may improve short-term mortality without increasing adverse events. Further research is needed to clarify the timing of IC measurements and target energy depending on this timing. However, this study marks a milestone that will direct future research toward investigating causal inferences for improving the outcomes of critically ill patients.

## Figures and Tables

**Figure 1 nutrients-16-01452-f001:**
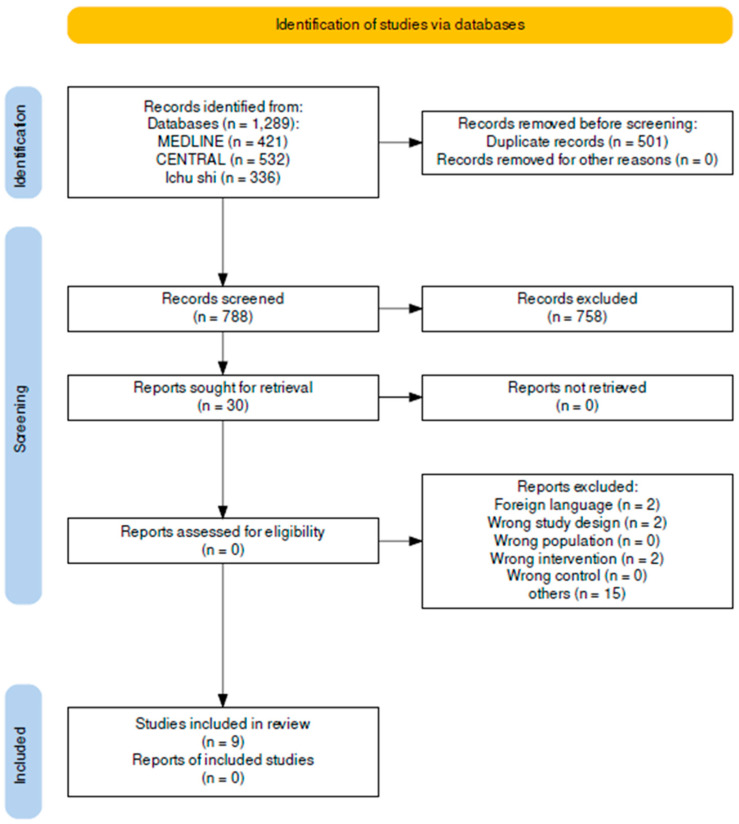
PRISMA 2020 flow diagram Ichushi; Igaku-Chuo-Zasshi.

**Table 1 nutrients-16-01452-t001:** Characteristics of included studies.

Author, Year, Country	Study Type	No. of Participants		Time to First Day of IC	Frequency	How to Administer Nutrition	Target Calories	REE (kcal)		Delivered Calories (Sufficiency Rate)	
		I	C					I	C	I	C
Jeffrey et al., 1990, USA [4]	Single-center RCT	26	23	Within 2 days	3 times/w	EN + PN	120%	2963	3781	3530 (120%)	3490 (94%)
Singer et al., 2011, Germany [32]	Single-center RCT	65	65	Within 2 days	Once every 2 days	EN + PN	100%	1976	1835	2086 (95%)	1480 (81%)
Anbar et al., 2014, Israel [33]	Single-center RCT	22	28	Within from 1 to 2 days	2 times	Oral intake + oral nutritional support	N/A	1274	1262	1121 (88%)	777 (61%)
Landes et al., 2016, USA [34]	Single-center RCT	15	12	Within 7 days	1 time/w	EN	110%	1976	2067	1709 (87%)	1592 (77%)
Allingstrup et al., 2017, Denmark [35]	Single-center RCT	100	99	As soon as possible	Once every 2 days	EN + PN	100%	2069	1875	1877 (91%)	1061 (57%)
Gonzalez-Granda et al., 2019, Germany [36]	Single-center RCT	20	20	Within from 1 to 3 days	3 times/w	EN + PN	Day 1: 25%, day 2: 50%, day 3: 75%, day 4: 100%	21.1/kg	25.0/kg	20.4/kg (98%)	20.0/kg (79%)
Azevedo et al., 2019, Brazil [37]	Single-center RCT	57	63	As soon as possible	Every day	EN + PN	N/A	1554	1450	1139 (73%)	1140 (79%)
Singer et al., 2020, Germany [5]	Multi-center RCT	209	208	Within from 1 to 2 days	Every day	EN + PN	80~100%	1953	1942	1746 (89%)	1301 (67%)
Farah et al., 2021, Malaysia [15]	Single-center RCT	73	73	Within from 1 day	Every day	EN + PN	70~100%	1512	1668	1507 (100%)	1519 (91%)

RCT, Randomized controlled trial; I, Intervention group; C, Control group; APACHE II, Acute Physiology and Chronic Health Evaluation II; N/A, not available; IC, Indirect calorimetry; ICU, Intensive care unit; EN, Enteral nutrition; PN, Parental nutrition; REE, Resting energy expenditure.

**Table 2 nutrients-16-01452-t002:** Summary of findings.

Outcomes	№ of Participants (Studies) Follow-Up	Certainty of Evidence (GRADE)	Anticipated Absolute Effects * (95% CI)		Relative Effect (95% CI)	Anticipated Absolute Effects	
			Risk Usual Care	Risk IC		Risk with Equation	Risk Difference with IC
Short-term mortality	988 (7 RCTs)	⨁⨁⨁◯ Moderate ^a^	25.7%	22.1% (18 to 27.2)	RR 0.86 (0.70 to 1.06)	257 per 1000	36 fewer per 1000 (77 fewer to 15 more)
Length of ICU stay	1090 (7 RCTs)	⨁⨁◯◯ Low ^b,c^	-	-	-		MD 0.86 higher (0.98 lower to 2.7 higher)
Duration of mechanical ventilation	1068 (7 RCTs)	⨁⨁⨁◯ Moderate ^c^	-	-	-		MD 0.66 higher (0.39 lower to 1.72 higher)
All infections	785 (4 RCTs)	⨁◯◯◯ Very low ^a,d^	22.1%	23.4% (18.1 to 30.2)	RR 0.99 (0.51 to 1.93)	221 per 1000	13 more per 1000 (40 fewer to 82 more)
Ventilator-associated pneumonia	785 (4 RCTs)	⨁⨁◯◯ Low ^a,b^	31.1%	11.5% (7.8 to 17.0)	RR 1.06 (0.49 to 2.28)	113 per 1000	2 more per 1000 (35 fewer to 58 more)
Physical functions (physical component summary)	309 (2 RCTs)	⨁⨁⨁◯ Moderate	-	-	**-**		MD 0.06 lower (6.28 lower to 6.15 higher)
Adverse events (kidney)	421 (2 RCTs)	⨁⨁⨁◯ Moderate ^a^	31.1%	32.0% (24.3 to 42.3)	RR 1.01 (0.77 to 1.34)	311 per 1000	9 more per 1000 (68 fewer to 112 more)
Adverse events (liver)	482 (2 RCTs)	⨁⨁⨁◯ Moderate ^a^	13.7%	13.7% (8.8 to 21.5)	RR 1.00 (0.64 to 1.57)	137 per 1000	0 fewer per 1000 (49 fewer to 78 more)

* The risk in the intervention group (and its 95% confidence interval) is based on the assumed risk in the comparison group and the relative effect of the intervention (and its 95% CI). CI: confidence interval; MD: mean difference; RR: risk ratio; IC: indirect calorimetric. Explanations ^a^. Downgraded one point for imprecision: because the sample size is less than N = 2000 (calculate OIS based on α = 0.05, β = 0.2, Event = 20%, RRR = 25%, N = 2000); ^b^. Downgraded one point for inconsistency: because the percentage of variation between studies (I2) is high; ^c^. Downgraded one point for imprecision: because the sample size is less than N = 800 (calculate OIS based on empirical thresholds; α = 0.05, β = 0.2, d = 0.2~0.3, N = 800); ^d^. Downgraded two points for inconsistency: because the percentage of variation between studies (I2) is high and significant in the heterogeneity test.

**Table 3 nutrients-16-01452-t003:** Post hoc analyses of primary outcomes.

Subgroup	IC Group n/Total (%)	Control Group n/Total (%)	RR (95% CI)
Short-term mortality (frequency of IC measurements)			
Every day	48/257 (18.6)	58/263 (22.1)	0.90 (0.66, 1.23)
Non-every day	59/233 (25.3)	70/235 (29.8)	0.86 (0.70, 1.06)
Short-term mortality (delivery calories)			
%REE < 90	48/279 (17.2)	60/291 (20.6)	0.89 (0.65, 1.21)
%REE ≥ 90	59/211 (28.0)	68/207 (32.9)	0.86 (0.70, 1.06)
Short-term mortality (mechanically ventilated patients)			
mechanically ventilated patients	104/442 (23.5)	124/447 (27.7)	0.78 (0.65, 1.21)
Non-mechanically ventilated patients	107/490 (21.8)	128/499 (25.7)	0.86 (0.70, 1.06)

IC, Indirect calorimetry; RR, Risk ratio; CI, Confidence interval; REE, Resting energy expenditure.

## Data Availability

All relevant data are within the manuscript and Appendix A.

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
