# Peer review of "Effects of Energy Delivery Guided by Indirect Calorimetry in Critically Ill Patients: A Systematic Review and Meta-Analysis"

_nutrients, 2024, doi:10.3390/nu16101452_

Round 1

Reviewer 1 Report

Comments and Suggestions for Authors

This is a systematic review to investigate the effect of  using IC to estimate energy expenditure in critically ill patients.

There are Nine RCTs involving 1178 patients included in this meta-analysis. 

The evidence obtained suggested that energy delivery by IC improved short-term mortality (risk ratio, 0.86; 95% confidence interval [CI], 0.70 to 1.06), but did not appear to affect the length of ICU stay (mean difference [MD], 0.86; 95% CI, -0.98 to 2.70) or the 

duration of mechanical ventilation (MD, 0.66; 95% CI, -0.39 to 1.72).

The authors concluded that use of IC may be of greater benefit for unstable, critically ill patients with diurnal fluctuations or daily changes in metabolic dynamics. The study topic is exciting, and the study design is robust. The conclusion is consistent with the result.

Author Response

  • Reviewer 1:

Comments and Suggestions for Authors

This is a systematic review to investigate the effect of using IC to estimate energy expenditure in critically ill patients.

There are Nine RCTs involving 1178 patients included in this meta-analysis. 

The evidence obtained suggested that energy delivery by IC improved short-term mortality (risk ratio, 0.86; 95% confidence interval [CI], 0.70 to 1.06), but did not appear to affect the length of ICU stay (mean difference [MD], 0.86; 95% CI, -0.98 to 2.70) or the duration of mechanical ventilation (MD, 0.66; 95% CI, -0.39 to 1.72).

The authors concluded that use of IC may be of greater benefit for unstable, critically ill patients with diurnal fluctuations or daily changes in metabolic dynamics. The study topic is exciting, and the study design is robust. The conclusion is consistent with the result.

RESPONSE: We greatly appreciate the thoughtful, insightful, and constructive comments from the editor. We would like to express our appreciation for taking the time to review our paper and provide critical inputs. We agree with the comments of reviewers 1. This study aims to clarify whether using IC to estimate energy expenditure in critically ill patients is more meaningful in improving survival rates than other estimation methods.

Reviewer 2 Report

Comments and Suggestions for Authors

Dear authors,

Thank you for giving me the opportunity to review your manuscript which, in an extensive paper, surveys the literature to establish the effects of indirect calorimetry-guided energy delivery in critically ill patients

The study is very interesting.

Here are my comments:

The INTRODUCTION chapter is very well written. The hypothesis and purpose are highlighted very well.

In the METHODS chapter, information is presented very well.

Please explain in more detail what it means for patients to leave intensive care sooner: their death or their recovery

RESULTS chapter

Figure 2 and Figure 3 should be uploaded to the additional materials section.

A table 3 would be necessary to systematize the results presented in figure 3.

The DISCUSSION chapter is well written.

I appreciate that the limitations of the article are well presented.

The CONCLUSIONS chapter does not analyze the impact of this study on the scientific and medical community.

The supplementary materials are very useful and representative.

Good luck!

Author Response

  • Reviewer 2:

Dear authors,

Thank you for giving me the opportunity to review your manuscript which, in an extensive paper, surveys the literature to establish the effects of indirect calorimetry-guided energy delivery in critically ill patients

The study is very interesting.

Here are my comments:

The INTRODUCTION chapter is very well written. The hypothesis and purpose are highlighted very well.

In the METHODS chapter, information is presented very well.

Please explain in more detail what it means for patients to leave intensive care sooner: their death or their recovery

Response

In response to reviewer 2's comment, we added to the Methods section that shorter ICU length of stay may be associated with worse mortality or improved disease status.

“while secondary outcomes included the following: (2) length of ICU stay, whose shorten-ing may be associated with worse mortality or improved disease status, and.”

Methods section; page 3, line 120-122.

RESULTS chapter

Figure 2 and Figure 3 should be uploaded to the additional materials section.

Response

Figures 2 and 3 have been moved to the Additional Materials section.

“Supplemental figure 2 and 3 “

A table 3 would be necessary to systematize the results presented in figure 3.

Response

Thank you very much for your suggestion. We created Table3 following Reviewer2's comment

“Table 3” Page 8

The DISCUSSION chapter is well written.

I appreciate that the limitations of the article are well presented.

The CONCLUSIONS chapter does not analyze the impact of this study on the scientific and medical community.

Response

Thank you for your important comments. We additionally described the impact of this study on directions for future research in the Conclusions section.

“However, this study marks a milestone that will direct future research toward investigating causal inferences for improving the outcomes of critically ill patients.”

Conclusion section; p 10, line 347-349

The supplementary materials are very useful and representative.

Good luck!

Reviewer 3 Report

Comments and Suggestions for Authors

This study is very important in field of clinical nutrition. I suggest few changes prior the acceptance for publication.

Abstract: ok

Introduction: What is hypothesis? Moreover, it is too short. 

Methods: This is very written. Well done.

Results: Several studies from dr Pichard wre not cited in discussed. Please, see: 

pichard c indirect calorimetry icu - Search Results - PubMed (nih.gov)

Discussion: To include Pichard´s papers. In addition, a limitation is the use of different IC equipament between the studies.

Conclusion: well written.

Author Response

  • Reviewer 3:

This study is very important in field of clinical nutrition. I suggest few changes prior the acceptance for publication.

Abstract: ok

Introduction: What is hypothesis? Moreover, it is too short. 

Responce

We agree with Reviewer 3's comment as a lack of description of the hypothesis of this study. In response to comments, we have added the research hypothesis to the Introduction section. Additionally, we have added an explanation of single-center RCTs reported since the previous SR.

“A single-center RCT in mechanically ventilated patients reported that nutritional therapy managed based on continuous IC measurements significantly reduced mean daily energy deficit and ICU mortality [15]. The hypothesis of this study is that using IC to estimate energy expenditure in critically ill patients will improve survival compared to other estimation methods..”

Introduction section: p2, line 72-76

Methods: This is very written. Well done.

Results: Several studies from dr Pichard wre not cited in discussed. Please, see: 

pichard c indirect calorimetry icu - Search Results - PubMed (nih.gov)

Discussion: To include Pichard´s papers. In addition, a limitation is the use of different IC equipament between the studies.

Responce

Thank you for your critical comment. We completely agree with reviewer 3's comment. Dr. Pichard's previous work mentioned about errors between IC devices, so we cited it in the discussion section [1], and we added the note that the use of the IC equipment, which varied between studies, might have influenced the results.

“Finally, previous studies mentioned errors due to differences in IC devices, and the use of different IC equipment in each study may have affected the results [40].”

Discussion section; p10, line 336-338

In addition, Dr. Pichard's previous work reported that the risk of nutrient overfeeding without the use of IC is a barrier to the administration of enteral nutrition and was therefore cited in the Discussion section [2].

“Berger et al reported that the risk of nutrient overdosage without IC creates a barrier to the administration of enteral nutrition [39].”

Discussion section; p9, line 320-321

References

  1. Berger, M. M.; Burgos, R.; Casaer, M. P.; De Robertis, E.; Delgado, J. C. L.; Fraipont, V.; Gonçalves-Pereira, J.; Pichard, C.; & Stoppe, C. Clinical nutrition issues in 2022: What is missing to trust supplemental parenteral nutrition (SPN) in ICU patients?. Critical care. 2022; 26(1), 271.
  2. Oshima, T.; Delsoglio, M.; Dupertuis, Y. M.; Singer, P.; De Waele, E.; Veraar, C.; Heidegger, C. P.; Wernermann, J.; Wischmeyer, P. E.; Berger, M. M.; & Pichard, C. The clinical evaluation of the new indirect calorimeter developed by the ICALIC project. Clinical nutrition. 2020; 39(10), 3105–3111.

Conclusion: well written.